# Biocalcification in porcelaneous foraminifera

**Zofia Dubicka[1,2,3]\*, Jarosław Tyszka[4], Agnieszka Pałczyńska[3], Michelle Höhne[1], Jelle Bijma[5], Max Jense[6], Nienke Klerks[6], Ulf Bickmeyer[1]**

[1]Ecological Chemistry, Alfred-Wegener-Institut Helmholtz-Zentrum für Polar- und Meeresforschung, Bremerhaven, Germany; [2]GFZ German Research Centre for Geosciences, Telegrafenberg, Potsdam, Germany; [3]Faculty of Geology, University of Warsaw, Warsaw, Poland; [4]Research Centre in Kraków, Institute of Geological Sciences, Polish Academy of Sciences, Kraków, Poland; [5]Marine Biogeosciences, Alfred-Wegener-Institut Helmholtz-Zentrum für Polar- und Meeresforschung, Bremerhaven, Germany; [6]Burgers' Ocean, Royal Burgers' Zoo, Arnhem, Netherlands

**\*For correspondence:**
z.dubicka@uw.edu.pl

**Competing interest:** The authors declare that no competing interests exist.

**Abstract** Living organisms control the formation of mineral skeletons and other structures through biomineralization. Major phylogenetic groups usually consistently follow a single biomineralization pathway. Foraminifera, which are very efficient marine calcifiers, making a substantial contribution to global carbonate production and global carbon sequestration, are regarded as an exception. This phylum has been commonly thought to follow two contrasting models of either *in situ* 'mineralization of extracellular matrix' attributed to hyaline rotaliid shells, or 'mineralization within intracellular vesicles' attributed to porcelaneous miliolid shells. Our previous results on rotaliids along with those on miliolids in this paper question such a wide divergence of biomineralization pathways within the same phylum of Foraminifera. We have found under a high-resolution scanning electron microscopy (SEM) that precipitation of high-Mg calcitic mesocrystals in porcelaneous shells takes place *in situ* and form a dense, chaotic meshwork of needle-like crystallites. We have not observed calcified needles that already precipitated in the transported vesicles, what challenges the previous model of miliolid mineralization. Hence, Foraminifera probably utilize less divergent calcification pathways, following the recently discovered biomineralization principles. Mesocrystalline chamber walls in both models are therefore most likely created by intravesicular accumulation of pre-formed liquid amorphous mineral phase deposited and crystallized within the extracellular organic matrix enclosed in a biologically controlled privileged space by active pseudopodial structures. Both calcification pathways evolved independently in the Paleozoic and are well conserved in two clades that represent different chamber formation modes.

## eLife assessment

This manuscript provides **important** information on the calcification process, especially the properties and formation of freshly formed tests (the foraminiferan shells), in the miliolid foraminiferan species Pseudolachlanella eburnea. The evidence from the high-quality SEM images is **convincing** although the fluorescence images only provide indirect support for the calcification process.

## Introduction

Over the past 500 million years, living organisms evolved different skeleton crystallization pathways. Very popular in nature is the mineralization of the extracellular matrix, e.g., in crustacean cuticles, mollusk shells, vertebrate bones, and teeth composed of dentin and enamel (*Weiner and Addadi, 2011*; *Kahil et al., 2021*; *Ujiié et al., 2023*). Radial foraminifera represented by rotaliids have been traditionally interpreted to make use of this crystallization mode (*Weiner and Addadi, 2011*). The

other two pathways are intravesicular and are characterized by either production of amorphous unstable phase within a large vesicle, such as a syncytium, well documented for sea urchin larvae (*Beniash et al., 1997*) or crystallization of calcite elements within smaller vesicles located in the intracellular space, as seen in fish that form guanine crystals and coccolithophores to produce coccoliths (*Weiner and Addadi, 2011*; *Kahil et al., 2021*). This model has also been attributed to the formation of porcelaneous shells by miliolid foraminifera (*Weiner and Addadi, 2011*) based on the model proposed by *Berthold, 1976*, and followed by .

As such, mineralization of shells in Foraminifera is believed to follow two highly contrasting pathways. The current theory states that Miliolida, characterized by imperforate, opaque milky-white shell walls (porcelaneous) (*Angell, 1980*; *de Nooijer et al., 2009*), produce fibrillar crystallites composed of Mg-rich calcite within tiny vesicles enclosed by cytoplasm. Miliolid shells are made of randomly distributed calcite needles that form a dense meshwork of chaotic crystallites that cause light reflection, resulting in opaque (porcelaneous) milky walls (*Hohenegger, 2009*). Calcite needles are thought to be precipitated completely within these vesicles and then transported to the site of chamber formation to be released via exocytosis (*Berthold, 1976*; *Angell, 1980*; *de Nooijer et al., 2009*; *de Nooijer et al., 2008*). The pre-formed needles or needle stacks are believed to be continuously embedded in an organic matrix (OM) in the shape of the new chamber until the wall is completed. Although this model is commonly accepted, it has never been sufficiently documented *in vivo*, and it does not resolve several conflicting issues. First of all, the question is how pre-formed bundles of parallel calcitic needles are transformed into randomly oriented needles within the shell wall. It is difficult to explain, if there is no recrystallization process within the wall structure after discharging the calcite crystallites. This problem was already emphasized by *Hemleben et al., 1987*. Second, why the newly constructed wall is still translucent after deposition of random crystals. We would expect a thin milky opaque layer of the new wall under normal transmitted light, as well as polarized crystals of calcite under crossed nicols. *Angell, 1980*, on his plate 2 presenting porcelaneous chamber formation in

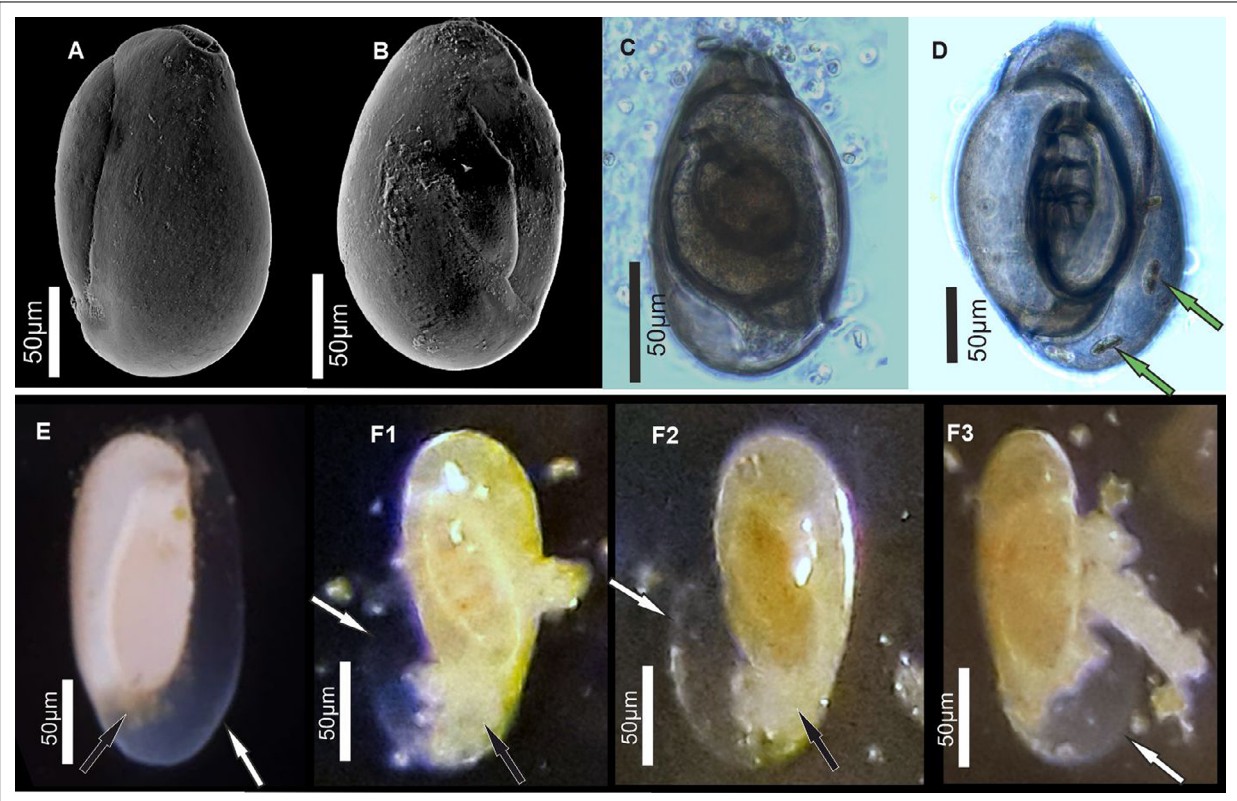

**Figure 1.** Specimens of miliolid foraminifera, identified as *P. eburnea* (d'Orbigny), used for experimental studies. (**A, B**) Scanning electron microscopy (SEM), (**C, D**) transmitted light microscope, and (**E, F**) stereomicroscope images. White arrows show the outer organic sheath of a new chamber during its gradual calcification expressed by its gradual appearance from complete transparency to milky and opaque aspect (**E, F**). Black arrows indicate a small mass of cytoplasm extruded from the aperture of the existing chamber. Green arrows point to incorporated algae.

miliolid *Spiroloculina hyalina* Schulze clearly documented the polarization front being shifted circa a half of the length of the new chamber behind the leading edge of the forming chamber. This shift represented more than an hour. Therefore, polarization was missing in the early and middle stage of chamber formation. It means that *Angell, 1980*, time lapse micrographs of the chamber formation were in conflict with the imaging under TEM. It seems that *Angell, 1980*, was aware of that problem and stressed that calcification had to be "intense enough to show under crossed nicols lags behind the leading edge of the forming chamber" (p. 93, pl. 2 figure 12/caption). In fact, all experiments that show the 'crystal vacuoles' (sensu *Angell, 1980*) documented under TEM (*Berthold, 1976*; *Angell, 1980*) required fixation of the samples, which was prone to post-fixation artifacts of unwanted calcite precipitation.

Our goal is to test whether the miliolid shell is produced by 'agglutination' of premade needle-like calcitic crystallites, and in consequence, whether this large group of calcareous Foraminifera follow crystallization within smaller vesicles located in the intracellular space. Therefore, we re-examined the mineralization process in Miliolida based on experiments on a living species, *Pseudolachlanella eburnea* (d'Orbigny) (*Figure 1*). This taxon was selected to facilitate replicated observations of chamber growth under controlled culture conditions. We included observations of *in vivo* biomineralization using multiphoton and confocal laser scanning microscopy (CLSM) followed by analyses of fixed specimens at different stages of chamber formation by high-resolution field emission scanning electron microscopy (FE-SEM) coupled with energy-dispersive X-ray spectrometry (EDS). Our new FE-SEM data challenge the current understanding of the biomineralization of miliolid foraminifera and such a significant divergence of biomineralization pathways within the Foraminifera.

## Results

All replicated *in vivo* experiments on *P. eburnea* facilitated by CLSM imaging with the application of membrane-impermeable Calcein and FM1-43 membrane dyes (performed in separate experiments) showed intravesicular fluorescence signals from groups of moving vesicles (1–5 µm in size) inside the cytosol (*Figure 2A and B*, *Figure 2—video 1* and *Figure 2—video 2*). The fluorescent vesicles inside the cytosol contained seawater, as documented by fluorescence of membrane-impermeable Calcein. These vesicles were taken up by endocytosis indicated by FM1-43 staining. This dye stains the cell membranes and indicates all endocytic vesicles by fluorescence, whereas the other intracellular vesicles remain unstained (*Amaral et al., 2011*). Both dyes demonstrate the uptake of seawater via the endocytosis of vesicles that are approximately 1–4 µm in diameter and move through the entire cell.

Additional LysoGlow84 staining revealed numerous acidic vesicles in the cytosol (*Figure 2C*, *Figure 2—video 3* and *Figure 2—video 4*). Acidic vesicles were accompanied by other vesicles (approximately 1–2 µm in size) that show autofluorescence upon multiphoton excitation at 405 nm (emission 420–480 nm), shown in red in *Figure 2C*. This wavelength partly permeates the shell to excite autofluorescence interpreted as associated with ACCs (see *Dubicka et al., 2023*). The autofluorescence of the shell itself is also present (*Figure 2D*), however, it is not clearly visible because the fluorescence of ACCs is much stronger. The intensity of the laser light is reduced because the multiphoton light has to pass through a thick three-dimensional carbonate wall of the foraminiferal shell. Further experimental studies are needed to confirm the ACC source of this autofluorescence and thus definitively eliminate potential organic sources of AF emissions.

In addition, typical chlorophyll autofluorescence (excitation at 405 or 633 nm, emission 650–700 nm, *Figure 2C*, *Figure 2—video 3* and *Figure 2—video 4* highlighted in green) was detected, which indicated the presence of chloroplasts in microalgae cells. These algal cells have been found to move within the cytosol of the observed specimens, in proximity of acidic vesicles and vesicles characterized by autofluorescence upon UV light (exc. 405 nm). These algal cells may represent facultative endosymbionts, as they were observed only during the chamber mineralization process in specimens with carbonate-bearing vesicles likely detected by *in vivo* CLSM experiments. They were documented just below the OM of the newly formed chamber, as seen in the FE-SEM observations as well as just below the OM of the newly created chamber as seen in the FE-SEM observations (*Figure 3—figure supplement 1G and H*). Specimens of *P. eburnea*, which displayed vesicles showing autofluorescence under UV light inside the cytosol, were fixed using Method B (see Materials and methods) coated with a few nanometers of carbon and analyzed by SEM-EDS. The main elements detected in the area of the fixed cytoplasm (*Figure 3—figure supplement 4*) were C, O, Na, Mg, P, S, Cl, K, and Ca (of particular

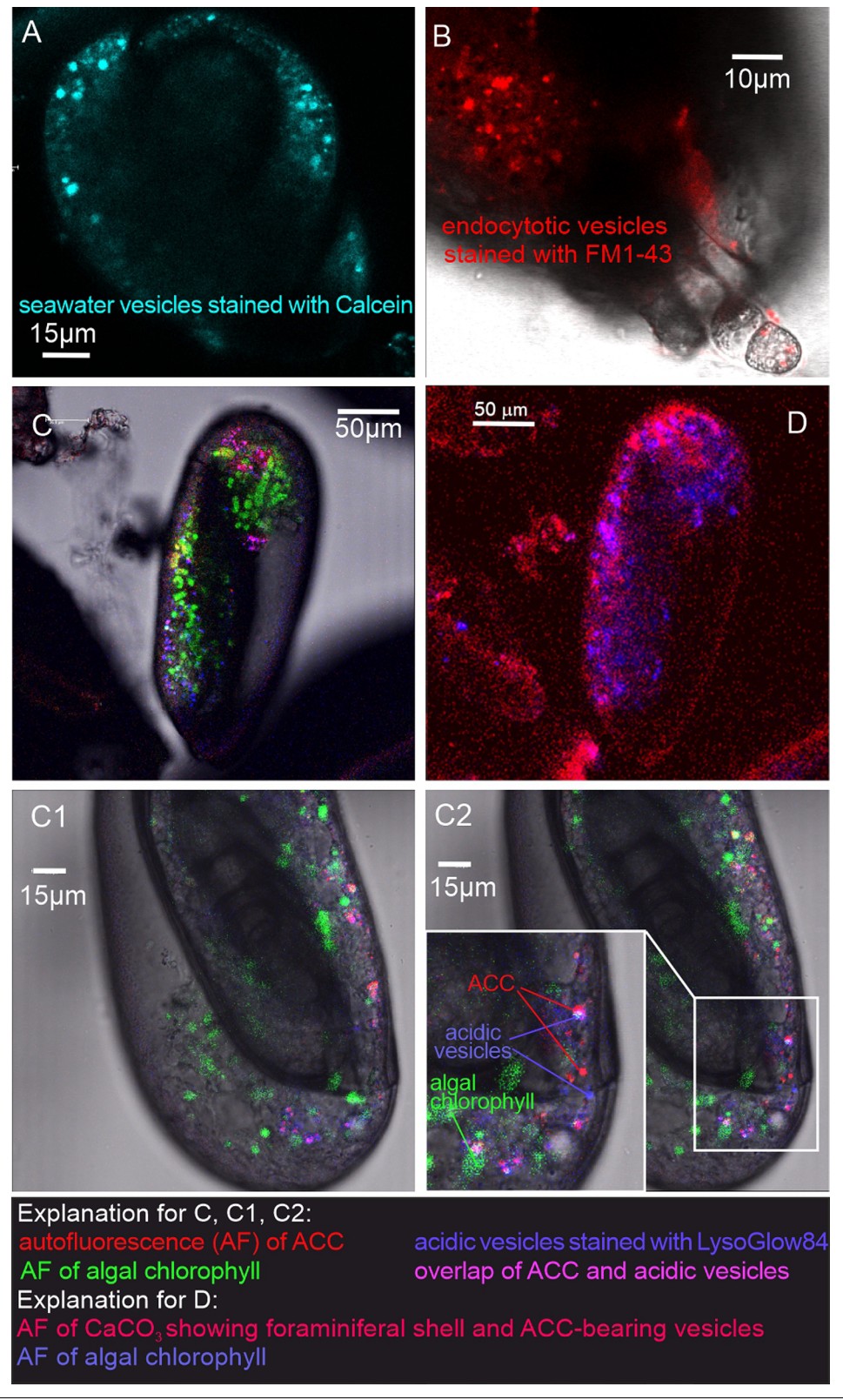

**Figure 2.** Fluorescence images of living *P. eburnea* conducted by confocal laser scanning microscopy. (**A**) Cell-impermeable Calcein (cyan) indicating endocytotic seawater vesicles, see *Figure 2—video 1*. (**B**) FM1-43 membrane dye indicating endocytotic vesicles (red), see *Figure 2—video 2*. (**C**) LysoGlow84 indicating acidic vesicles (navy blue), autofluorescence of chloroplasts (green), and Mg-ACC pools (red), see *Figure 2—videos 3*

*Figure 2 continued on next page*

*Figure 2 continued*

*and 4* (note the overlap of ACC and acidic vesicles is marked in lilac). (**D**) Autofluorescence image with reduced threshold of the studied Miliolida species (exc. 405 nm) showing algal chlorophyll (blue) and $CaCO_3$ (red), both ACC and calcite shell.

The online version of this article includes the following video(s) for figure 2:

**Figure 2—video 1.** Living *P. eburnea* showing cell-impermeable Calcein (blue, exc. 488 nm, em. 505–555) in a series of 107 overlaid images taken during 428 s.

https://elifesciences.org/articles/91568/figures#fig2video1

**Figure 2—video 2.** FM1-43 membrane probe fluorescent signals (red, exc. 488 nm, em. 580–620 nm) emitted by intracellular vesicles within cytosol of *P. eburnea*.

https://elifesciences.org/articles/91568/figures#fig2video2

**Figure 2—video 3.** Living *P. eburnea* showing fluorescence signal inside the cytosol: autofluorescence of Mg-ACC pools (red, exc. 405 nm, em. 420–490 nm) and algal chloroplasts (green, exc. 633 nm, em. 640–690 nm), fluorescent signal of LysoGlow84 pH-sensitive dye (exc. MP 720 nm, em. 440–470 nm) indicating acidic vesicles.

https://elifesciences.org/articles/91568/figures#fig2video3

**Figure 2—video 4.** Living *P. eburnea* showing fluorescence signal inside the cytosol: autofluorescence of Mg-ACC pools (red, exc. 405 nm, em. 420–490 nm) and algal chloroplasts (green, exc. 633 nm, em. 640–690 nm), fluorescent signal of LysoGlow84 pH-sensitive dye (exc. MP 720 nm, em. 440–470 nm) indicating acidic vesicles.

https://elifesciences.org/articles/91568/figures#fig2video4

interest were the high contents of Mg and Ca), whereas the main elements detected within the area of the new chamber in the form of a gel-like matter filled with dispersed nanograins were C, O, Na, Mg, S, Cl, and Ca (*Figure 3—figure supplement 4*). The shell content was strongly enriched with Ca relative to the cytoplasm, which showed a much higher Mg/Ca ratio.

FE-SEM observations of the fully mineralized test walls displayed the porcelaneous structures (see *Parker, 2017*; *Dubicka et al., 2018*), which are made of three mineralized zones, i.e., (1) extrados that represents an outer mineralized surface (approximately 200–300 nm in thickness; *Figure 3—figure supplements 1C and 2C*); (2) porcelain that denotes the main body of the wall constructed from randomly oriented needle-shaped crystals (up to 1–2 µm in length and approximately 0.2 µm in width). No gel-like matter was observed between the needles of the porcelain structures that appeared in the early stages of wall formation (*Figure 3E, E1*; *Figure 3—figure supplements 2C and 3A*); and (3) intrados that represents an inner mineralized surface (approximately 200–300 nm in thickness) made of needle-shaped crystallites (*Figure 3E, E1* and *Figure 3—figure supplement 1A*).

Growing chambers, captured at the various successive stages of chamber formation in different specimens, have revealed the following morphological features: (1) a solitary, thin organic sheath (approximately 200–300 nm thick) that represents the most distal part of the new chamber and is anchored to the older, underlying solid calcified chamber (*Figure 4A*); (2) a solitary, outer organic sheath (OOS) filled with spread calcifying nanograins (*Figure 4B*; *Figure 3—figure supplement 2*); (3) a gel-like matter (4–5 µm in thickness) with a granular texture, bounded on two sides by intrados and extrados, and containing relatively widely spaced, randomly dispersed carbonate nanograins (*Figure 3A, B*; *Figure 4C*; *Figure 3—figure supplement 1A–D*); (4) the test inside made of chaotic meshwork of carbonate nanograins partly transformed to short needles with a small amount of gel-like OM in-between (*Figures 3C, D and 4D*); (5) the test inside composed of needle-shaped crystals with planar faces and no apparent remaining gel-like matter (*Figures 3E, E1 and 4E*). Carbonate nanograins at the shell construction site were well documented in our SEM-EDS studies (*Figure 3—figure supplement 4*). Both fixation methods (see Materials and methods) yielded highly consistent results.

## Discussion
### Porcelaneous shell formation

Comparative analysis of the nanostructures of the newly built chambers combined with the elemental composition obtained from SEM-EDS, as well as the data from CLSM, allowed us to identify important steps in the accretive formation of *P. eburnea* shells. The formation of a new chamber

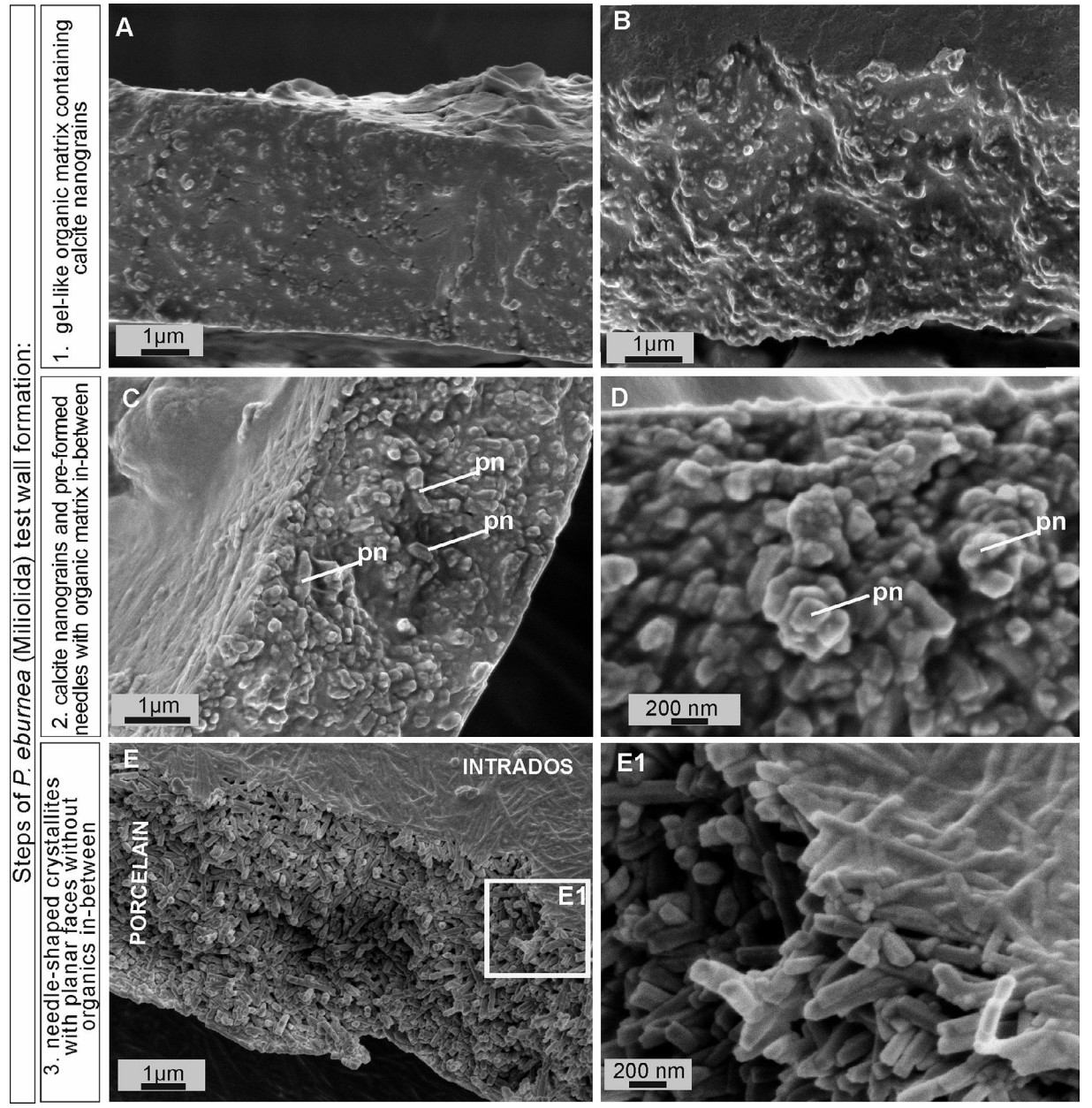

**Figure 3.** Scanning electron microscopy (SEM) images of the major steps of the formation of *P. eburnea* shell-building components. Test cross-section showing: (**A, B**) carbonate nanograins within organic matrix, (**C, D**) nanograins merging into needle-like mesocrystals, (**E**) fully developed needle-shaped elements; pn – nanograins partly transformed to short needles.

The online version of this article includes the following source data and figure supplement(s) for figure 3:

**Source data 1.** SEM images of fixed *P. eburnea*.

**Figure supplement 1.** Scanning electron microscopy (SEM) images of the new shale formation site of *P. eburnea*.

**Figure supplement 1—source data 1.** SEM images of fixed *P. eburnea*.

**Figure supplement 2.** Scanning electron microscopy (SEM) images of fixed *P. eburnea* during calcification process.

**Figure supplement 2—source data 1.** SEM images of fixed *P. eburnea*.

**Figure supplement 3.** Scanning electron microscopy (SEM) images of newly built chamber and previous chambers of one specimen of *P. eburnea*.

**Figure supplement 3—source data 1.** SEM images of *P. eburnea*.

**Figure supplement 4.** Energy-dispersive X-ray spectrometry (EDS) analysis.

*Figure 3 continued on next page*

*Figure 3 continued*

**Figure supplement 5.** Scanning electron microscopy (SEM) images of miliolid *Agathamina pusilla* Geinitz from the Lower Permian (ca. 290 Mya) of the Holy Cross Mountains (Poland) showing needle test structure identical to that of Recent taxa.

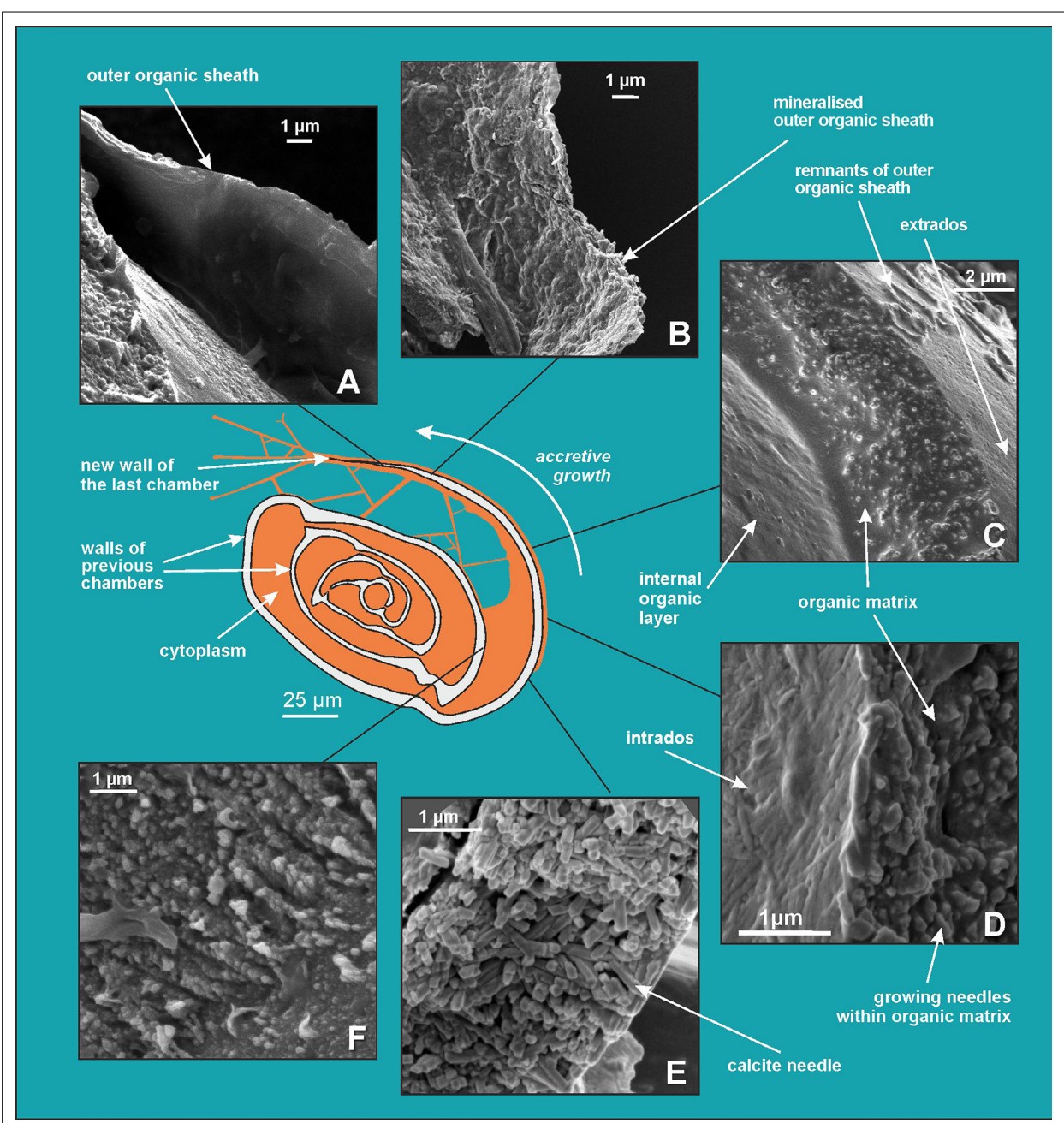

**Figure 4.** Scanning electron microscopy (SEM) images showing successive stages of new chamber formation in *P. eburnea*. (**A**) Outer organic sheath, (**B**) mineralized outer organic sheath, (**C**) calcite nanograins within a gel-like organic matrix, (**D**) needle-shaped mesocrystal growth, (**E**) needle-like calcite building elements, (**F**) nanogranular shell (interval view).

The online version of this article includes the following source data for figure 4:

**Source data 1.** SEM images of *P. eburnea*.

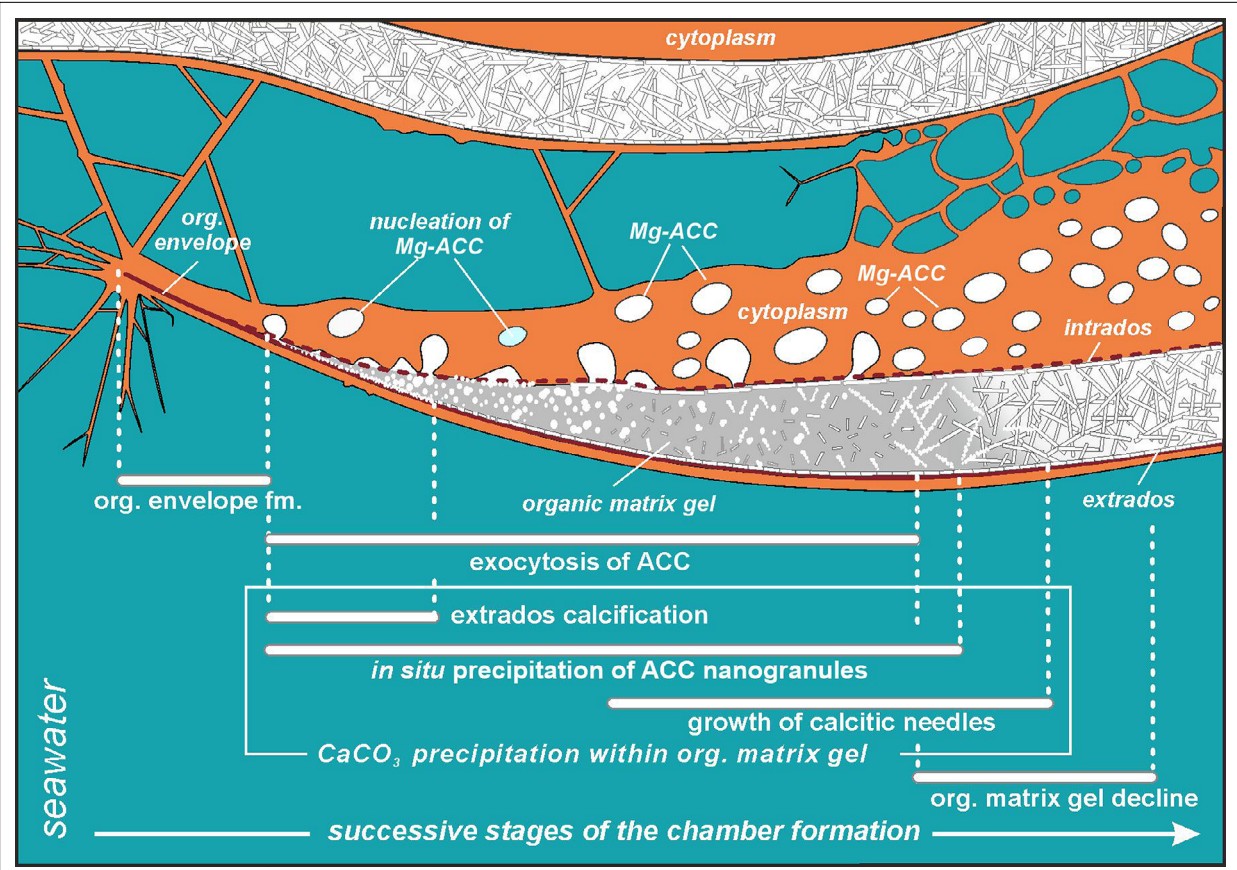

**Figure 5.** Simplified model of porcelaneous wall construction based on foraminifer *P. eburnea*. White spots labeled as Mg-ACC represent vesicles with Mg-rich amorphous calcium carbonates.

begins with the construction of a thin OOS that pre-shapes the new chamber (*Figures 4A and 5*). The OOS is made by pseudopodial structures supported by the cytoskeleton immediately after the extrusion of a small mass of cytoplasm from the aperture (*Figure 1E and F*). Once the OOS is constructed, the first calcium carbonate accumulation takes place inside in the form of carbonate nanograins (*Figures 4B and 5*, *Figure 3—figure supplement 2A and B*), creating the extrados. The extrados stabilizes the final chamber morphology relatively quickly. Subsequently, the wall gradually thickens through the primary accumulation of hydrated and amorphous Mg-rich $CaCO_3$ (*Figures 4B and 5*). We suppose that the carbonate content is successively deposited by exocytosis of Mg-ACC-rich vesicles that most likely represent the vesicles converted from seawater stained with Calcein (*Figure 5*). The characteristic autofluorescence inside foraminiferal cell excited at 405 nm (*Figure 2*; *Figure 2—video 3* and *Figure 2—video 4*) most likely indicates the carbonate content of the vesicles, which are considered to be Mg-ACCs (see *Dubicka et al., 2023*). Mg-ACC is an unstable, amorphous and hydrated form of $CaCO_3$ with a significantly high concentration of Mg (*Raz et al., 2000*; *Weiner et al., 2003*; *Bentov and Erez, 2006*; *Kahil et al., 2021*) and is commonly regarded as a resource for most biocalcification processes. ACCs have been found in many calcifying marine organisms, such as echinoderms, mollusks, coccolithophorid algae, cyanobacteria, crustaceans, and rotaliid foraminifera, where they are typically interpreted as pre-material phases for the production of calcite skeletons (*Hasse et al., 2000*; *Weiss et al., 2002*; *Sviben et al., 2016*; *Dubicka et al., 2018*; *Kahil et al., 2021*). Research suggests that a high-Mg content not only makes ACC unstable but also facilitates the transport of ACC to the crystallization site, where it is initially transformed into carbonate nanograins (*Cölfen and Qi, 2001*; *Addadi et al., 2003*; *Raz et al., 2003*; *Dubicka et al., 2023*). The existence of intracellular, vesicular intermediate amorphous phase (Mg-ACC pools), which supplies successive doses of carbonate material to shell production, might be supported not only by autofluorescence (excitation at 405 nm; *Figure 2*; *Figure 2—video 3* and

*Figure 2—video 4*; see *Dubicka et al., 2023*) but also by a high content of Ca and Mg analyzed in the cytoplasmic area by SEM-EDS analysis (*Figure 3—figure supplement 4*). In the future, more precise higher resolution elemental measurements are needed for better documentation of miliolid ACC-bearing vesicles. However, the small size of carbonate-bearing vesicles (approximately 1–2 µm) may make this difficult.

$Mg^{2+}$ and $Ca^{2+}$ ions for intravesicular production of Mg-ACCs are obtained from seawater and taken up by endocytosis, as independently indicated by membrane-impermeable Calcein, as well as by the FM1-43 probe selectively labeling membranes of endocytic vesicles (*Figure 2A and B*, *Figure 2—video 1* and *Figure 2—video 2*). We hypothesize that vesicles are carried along cytoskeletal structures to the OM, as observed in rotaliid foraminifera (*Dubicka et al., 2023*), where they dock and release their contents (*Figure 4*). The nanograins then precipitate within the gelatinous matter that consists of amorphous carbonates and OM released from the vesicles (*Figure 3A–C*; *Figure 4C*; *Figure 5*). Nanograins immersed in the gel-like matter gradually grow into needle-shaped elements, precipitating *in situ* within the final wall structure (*Figure 3C, D*; *Figure 4D*). The gel-like matter appears to be involved in needle formation; however, the OM seems to disappear (*Figures 3E and 4E*) when the needle-shaped crystals are created. We suspect that the gel-like matter consists of preformed liquid amorphous mineral phase (Mg-ACC) within the extracellular OM that is suggested by the EDS spectra of the early stage of the wall calcification (*Figure 3—figure supplement 4*: A3 area). The calcification of extrados and intrados occurs before the interior of the wall crystallizes, providing stability to the new chamber at both edges of the wall (*Figure 4D*).

The protruding cytoplasm appears to immediately form a chamber wall by secreting OM and crystals from the vesicles (*Angell, 1980*). As calcite secretion continues along the leading edge, the newly formed segment remains covered by a thin, moving sheet of cytoplasm that is called by *Angell, 1980*, the 'active sheet'. This thin active sheet of cytoplasm may represent a lamellipodium that is a pseudopodial structure known to be involved in the biomineralization of Rotaliida (*Tyszka et al., 2019*). It is also likely that reticulopodial structures (that do not coat the whole calcification site) are responsible for the distribution and shape of the internal surface of the chamber wall. That occurs by successive accumulation of ACC and OM as identified on TEM images by *Angell, 1980*. His results suggest that crystallization of calcite needles is 'limited to a confined space controlled by active cytoplasmic structures' that are strictly separated by the membranes from the cytosol.

## Formation of shell crystallites: a paradigm shift

Miliolids were thought to share a similar, intracellular, crystallization pathway as the coccolith formation in coccolithophorids (*Weiner and Addadi, 2011*) that evolved in the Triassic, i.e., ca. 210 Myr ago (*Gardin et al., 2012*). Coccoliths are produced within intracellular Golgi-derived vesicles and then exported to the surface of the extracellular coccosphere (*Borman et al., 1986*). Miliolids, with their unique fibrillar calcitic microstructures, evolved much earlier, i.e., ca. 300 Myr ago in the late Paleozoic (*Figure 3—figure supplement 5*). Until now, it was generally considered that calcite crystals in miliolids also precipitate within vesicles immersed in the cytoplasm and are then transported to the location of the wall construction, where they are released by exocytosis (*Berthold, 1976*; *Angell, 1980*; *de Nooijer et al., 2009*; *Weiner and Addadi, 2011*). Our FE-SEM study of *P. eburnea* shows the lack of premade needle-like crystallites of calcite at the early stages (I–IV) of the wall formation. In contrast, we can clearly infer the *in situ* calcification front with a progressive sequence of crystal growth behind the leading edge of the forming chamber (*Figures 4 and 5*). Therefore, this miliolid species apparently does not produce shells by 'agglutination' of premade needle-like crystallites of calcite, in contrast to the traditional miliolid calcification model (*Berthold, 1976*; *Angell, 1980*).

In the light of these results, another argument emerges that further confirms *in situ* calcification of miliolid chambers. It explains the extended transparency of unmineralized walls observed under the light stereomicroscope. The chamber wall under formation tends to gradually change its appearance during calcification from completely transparent to milky and opaque (*Figure 1E and F*).

Our results on biomineralization of this miliolid species do not confirm the formation of individual skeletal crystallites within intracellular vesicles. However, in turn, our results do support existence of endocytotic vacuolization of seawater in miliolids that was first suggested by . We further support *Angell, 1980*, interpretation that the calcite crystals are dispersed in the gel-like OM (see *Figure 3A, B*; *Figure 4C, D*; *Figure 5*). This gel-like fluidal OM likely includes a rich Mg-ACC component as the

substrate for *in situ* calcification (*Figures 3–5*). Interestingly, the previous studies by *Angell, 1980*, did not support crystal formation within vacuoles either.

Precipitation of calcite nanograins, which then merge and transform into crystallites, probably occurs within the organic matter after the release of $Mg^{2+}$ from Mg-ACC. The organic matter provides an appropriate physiochemical microenvironment for initiating and maintaining the crystallization process by manipulating many essential factors, including pH, and kinetics of the system (*Kahil et al., 2021*). According to *Tyszka et al., 2021*, the OM involved in the biomineralization of foraminiferal miliolid shells may contain collagen-like networks.

Our *in vivo* CLSM observations show a miliolid cytoplasm containing intracellular carbonate-bearing vesicles. Such vesicles have been well documented by *Angell, 1980*, who stressed their crucial role in the biomineralization process. However, rather than transporting pre-formed solid needles, the vesicles likely carry liquid or quasi-liquid calcification substrates. This liquid carbonate phase was apparently maintained by a relatively high concentration of Mg (*Figure 3—figure supplement 4*), which was much higher than that in the shell, as detected by the SEM-EDS analyses.

Recently, an independent study was performed on another miliolid species – *Sorites orbiculus* (*Nagai et al., 2023*). The researchers reported highly complementary results that indicate the lack of crystal-like structures within the intracellular vesicles. Their results suggested that calcification of this miliolid species did not follow *Hemleben et al., 1987* model because intracellular vesicles did not produce needle-like crystals to construct the shell wall. They also stated that their observations 'may reveal a novel and unknown mode of biomineralization in foraminifera'.

Because, miliolid wall texture originated together with the appearance of miliolid foraminifera as it has also been recorded within Paleozoic taxa (*Figure 3—figure supplement 5*), thus the calcification mode of miliolids apparently evolved in the late Paleozoic (≥350 Mya) and is well conserved in this clade till today. It should be emphasized that our recent understanding of all calcification pathways in Foraminifera implies their independent evolution within main phylogenetic groups, besides miliolids and rotaliids, also including spirillinids, nodosariids, and robertinids (*Pawlowski et al., 2013*; *Mikhalevich, 2013*; *Dubicka et al., 2018*; *Dubicka, 2019*; *Sierra et al., 2022*; *de Nooijer et al., 2023*). In fact, most of these biomineralization evolutionary transitions from agglutination to calcification originated in the mid- and late Paleozoic.

Mg-ACC has also recently been documented in rotaliid foraminifera (*Mor Khalifa et al., 2018*; *Dubicka et al., 2023*). Therefore, the biocalcification processes in Rotaliida and Miliolida, which belong to the two main foraminiferal classes Globothalamea and Tubothalamea, respectively (*Pawlowski et al., 2013*), are more alike than previously thought (*Weiner and Addadi, 2011*). Their mesocrystalline chamber walls are created by accumulating and assembling nanoparticles of pre-formed liquid amorphous mineral phase. Their calcification occurs within the extracellular OM enclosed in a biologically controlled privileged space by active pseudopodial structures. However, we are aware that this process must also vary to some extent as the chemical composition of the calcite, as well as primary crystallite geometries differ between the groups. Seawater provides the relevant Ca and Mg ions for calcification, which are taken up in both groups by endocytosis. In *Amphistegina* (Rotaliida), this process is performed by shell pores (*Dubicka et al., 2023*), as well as aperture; in non-porous Miliolida, it is done by granuloreticulopodia emanating from the aperture (*Figure 2*, *Figure 2—video 2*). In both the rotaliid *Amphistegina* and the miliolid *P. eburnea* carbonate-bearing vesicles are surrounded by moving acidic vesicles (*Figure 2*, *Figure 2—video 3* and *Figure 2—video 4*), which likely facilitate pH regulation at the mineralization front (see *Toyofuku et al., 2017*; *Chang et al., 2023*). It is very likely that pH is controlled by active outward proton pumping by a V-type H+ ATPase or proton outflux driven by pH that is responsible for the proton flux and related calcification (*Toyofuku et al., 2017*; see also *Matt et al., 2022*). We suspect much higher pH values within vesicles transporting Mg-ACC to the site of calcification. Such alkaline vesicles were detected by the HPTS fluorescent labeling and reported by several previous studies (*de Nooijer et al., 2008*; *de Nooijer et al., 2009*).

Our findings are in line with recent work in biomineralization, supporting that 'biominerals grow by the accretion of amorphous particles, which are later transformed into the corresponding mineral phase' (*Macías-Sánchez et al., 2011*; see also *Meldrum and Cölfen, 2008*). Miliolid needles, assembled with calcite nanoparticles, are unique examples of biogenic mesocrystals (see *Cölfen and Antonietti, 2005*), as they form distinct geometric shapes limited by planar crystalline faces. Mesocrystals are constructed from highly ordered individual nanoparticles (*Cölfen and Mann, 2003*; *Sturm (née*

*Rosseeva) and Cölfen, 2016*; *Sturm (née Rosseeva) and Cölfen, 2017*) that form hierarchically structured solid materials in the crystallographic register and are rather devoid of outer planar surfaces. These result from the aggregation, self-assembly, and mesoscopic transformation of amorphous precursor nanoparticles. Mesocrystals are common biogenic components in the skeletons of marine organisms, such as corals, echinoderms, bivalves, sea urchins, and rotaliid foraminifera (e.g. *Macías-Sánchez et al., 2011*; *Benzerara et al., 2011*; *Seto et al., 2012*; *Evans, 2019*; *Dubicka et al., 2023*).

Our biomineralization model further explains the random orientation pattern of the calcite needles within the shell wall. The miliolid intertwined calcitic structure cannot be explained by the models proposed by *Berthold, 1976*, and followed by , i.e., by the successive deposition of vesicles with ready bundles of solid calcitic fibers (needles) without additional recrystallization processes. In our proposed *in situ* calcification model, calcite crystallites have sufficient space to grow within the flexible gelatinous OM. In addition, our model explains the need for a light and dark phase for the algae that are present inside *P. eburnea* during the biomineralization processes, as these algae possibly play an important role. Small miliolid coiling foraminifera has been regarded as a non-symbiotic taxon because their shells are not transparent, however, this is not true for red and infrared light. Fully developed miliolid shells are made of randomly distributed needles that cause light reflection, resulting in opaque (porcelaneous) walls that possibly protect the foraminifera from UV irradiation and allow them to live in extremely illuminated shallow seas (*Hohenegger, 2009*). These walls are permeable to red and infrared light, as we observed using multiphoton laser. Red light is commonly believed to be the most efficient waveband for photosynthesis, however green light may achieve higher quantum yield of $CO_2$ assimilation and net $CO_2$ assimilation rate (*Liu and van Iersel, 2021*). *P. eburnea* may acquire its facultative symbionts only for the duration of the biomineralization process. The late stage of needle formation in the shell production process ensures that the wall remains transparent by the time the needles are completed. Similar patterns of the gradual change from transparent to opaque whitish walls were also observed in larger symbiotic miliolids by *Marshalek, 1969*, *Wetmore, 1999*, and *Tremblin et al., 2021*. The latter authors (*Tremblin et al., 2021*) documented chamber formation of miliolid *Vertebralina striata* with cytoplasm enveloped by a transparent sheath decorated with striate already present in the transparent wall before calcification. They also interpret white areas on the sheath, indicating incipient concentrations of minute calcite crystallites that represent the mineralized wall. The biomineralization process is likely aided by their dark respiratory activity (see *Hallock, 1999*), as they could supply calcification substrates such as $HCO_3^-$ through respiration or by increasing pH at the calcification site during the light phase. Similarly, representatives of miliolid large benthic foraminifera (Archaiasidae, Soritidae, and Peneroplidae) host endosymbiotic algae (*Lee, 2006*; *Prazeres and Renema, 2019*). Therefore, they have developed additional morphological and textural features such as pits, grooves/striate, or windows, which enable light penetration into the places where symbionts are positioned (see *Hohenegger, 2009*; *Parker, 2017*).

## Materials and methods

Living foraminifera, collected from the coral reef aquarium in the Burgers' Zoo (Arnhem, Netherlands), were cultured in a 10 L aquarium containing seawater with a salinity of 32 psu, pH of 8.2, and a

**Table 1.** Wavelengths and dyes.

| Dye | Concentration | Excitation nm | Emission nm | Source | Function |
|---|---|---|---|---|---|
| LysoGlow84 | 50 µM | Multiphoton 730 | 380–415/450–470 | Marnas Biochemicals | pH, membrane permeable |
| FM1-43 | 1 µM | Argon 488 or Multiphoton 1000 nm | 580–620 | Thermo Fisher Scientific | Membrane staining |
| Calcein | 0,7 mg/10 mL | Argon 488 | 510–555 | Thermo Fisher Scientific | Membrane-impermeable water soluble dye |
| Autofluorescence | Diode 405 MP 800 | 420–490 | | | $CaCO_3$, ACC |
| Autofluorescence | Diode 405/HeNe 633 | 650–700 | | | Chlorophyll of algae |

temperature of 24°C. Specimens of *P. eburnea* (d'Orbigny) were placed in 4 mL Petri dishes 1 day prior to CLSM studies, incubated without food for 18–24 hr, and then individuals that underwent chamber formation were observed under a Zeiss Stemi SV8 stereomikroscope for selection of individuals.

These selected individuals were studied *in vivo* using a Leica SP5 Confocal Laser Scanning Microscope equipped with an argon, helium-neon, neon, diode, and multiphoton Mai Tai laser (Spectra-Physics) at the Alfred-Wegener-Institut, Bremerhaven, Germany. *In vivo* experiments were performed by labeling samples with different fluorescent dyes (*Table 1*) just before imaging using pH-sensitive LysoGlow84 (50 µM exc. MP 720 nm exc./em. 380–415 nm and 450–470 nm, Marnas Biochemicals Bremerhaven, incubation time: 2 hr), FM1-43 membrane stain (1 µM, exc. 488 nm em. 580–620 nm, Invitrogen, incubation time: 24 hr), and membrane-impermeable Calcein (0.7 mg/10 mL, exc. 488 nm, em. 510–555 nm, incubation time: 24 hr). The foraminifera were removed from the Petri dish with clean water using a pipette. In addition, the autofluorescence of specific foraminiferal structures at the chosen excitation/emission wavelength was detected. All experiments were replicated with at least several individuals of the same species. All fluorescence probe experiments were performed with appropriate controls.

Additional foraminifera individuals that had been studied by CLSM were fixed for further analysis. The fixation process followed two different methods: (1) 60 individuals were transferred to 3% glutaraldehyde for 5 s and then dehydrated stepwise for a few seconds with an ethanol/distilled water mixture with increasing concentrations (30%, 50%, 70%, and 99%). (2) The seawater was removed from 50 individuals by pipetting and applying a small piece of Kimtech lab wipe (without any rinsing), followed by quick drying in warm air (30–35°C). This method stops the dissolution of the amorphous mineral phase because there is no contact with other liquids. Fixed foraminifers of both procedures were gently broken using a fine needle to coat the cross-sectional surfaces and tested inside with a few nanometers of either gold or carbon. Foraminifera were then studied using a Zeiss Sigma variable-pressure FE-SEM equipped with EDS at the Faculty of Geology, University of Warsaw.

## Acknowledgements

This work was supported by the Alexander von Humboldt Foundation Research Fellowship for experienced researchers to ZD and the Polish National Science Center (UMO-2018/29/B/ST10/01811) to JT and ZD, a grant coordinated by Grzegorz Racki, University of Silesia. We kindly thank Oscar Branson for his comments on an earlier version of the manuscript and the suggested improvements. We sincerely appreciate the valuable feedback and constructive criticism provided by the anonymous reviewers.

## Additional information

### Funding

| Funder | Grant reference number | Author |
| --- | --- | --- |
| Alexander von Humboldt-Stiftung | | Zofia Dubicka |
| Polish National Science Center | UMO-2018/29/B/ST10/01811 | Jarosław Tyszka Zofia Dubicka |

The funders had no role in study design, data collection and interpretation, or the decision to submit the work for publication.

### Author contributions

Zofia Dubicka, Conceptualization, Formal analysis, Funding acquisition, Investigation, Methodology, Writing - original draft; Jarosław Tyszka, Conceptualization, Methodology, Writing - original draft; Agnieszka Pałczyńska, Investigation, Writing – review and editing; Michelle Höhne, Investigation; Jelle Bijma, Writing – review and editing; Max Jense, Writing – review and editing, Provide living foraminifera; Nienke Klerks, Provide living foraminifera; Ulf Bickmeyer, Conceptualization, Formal analysis, Investigation, Methodology, Writing – review and editing

## Author ORCIDs

Zofia Dubicka  https://orcid.org/0000-0003-1105-4111
Jarosław Tyszka  http://orcid.org/0000-0002-1630-3415
Nienke Klerks  https://orcid.org/0000-0002-9285-9976
Ulf Bickmeyer  http://orcid.org/0000-0002-5351-2902

Reviewer #1 (Public Review): https://doi.org/10.7554/eLife.91568.3.sa1
Reviewer #2 (Public Review): https://doi.org/10.7554/eLife.91568.3.sa2
Author response https://doi.org/10.7554/eLife.91568.3.sa3

## Additional files

### Supplementary files

- MDAR checklist

### Data availability

All data generated or analysed during this study are included in the manuscript and supporting files.

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
