## [Editor Report · eLife assessment]

This manuscript provides **important** information on the calcification process, especially the properties and formation of freshly formed tests (the foraminiferan shells), in the miliolid foraminiferan species Pseudolachlanella eburnea. The evidence from the high-quality SEM images is **convincing** although the fluorescence images only provide indirect support for the calcification process.

---

## [Referee Report · Reviewer #1 (Public Review)]

Summary:

The manuscript by Dubicka and co-workers on calcification in miliolid foraminifera presents an interesting piece of work. The study uses confocal and electron microscopy to show that the traditional picture of calcification in porcelaneous foraminifera is incorrect.

Strengths:

The authors present high-quality images and an original approach to a relatively solid (so I thought) model of calcification.

Weaknesses:

There are several major shortcomings. Despite the interesting subject and the wonderful images, the conclusions of this manuscript are simply not supported at all by the results. The fluorescent images may not have any relation to the process of calcification and should therefore not be part of this manuscript. The SEM images, however, do point to an outdated idea of miliolid calcification. I think the manuscript would be much stronger with the focus on the SEM images and with the speculation of the physiological processes greatly reduced.

Comments on revised version:

I continue to disagree. As the authors acknowledge: 'may be a hint indicating ACC...', but it may also be something else. This is really something else than showing ACC is involved in foraminiferal calcification. I still think the reasoning is shaky and below, I will clarify why the fluorescence may well not be related to ACC and in fact, some or even most of the vesicles may not play the role that the authors suggest. Even if they do, the conclusions are not supported by the data presented here. Unfortunately, I found some of the other answers to my question not satisfactory either.

---

## [Referee Report · Reviewer #2 (Public Review)]

Summary:

Dubicka et al. in their paper entitled " Biocalcification in porcelaneous foraminifera" suggest that in contrast to the traditionally claimed two different modes of test calcification by rotallid and porcelaneous miliolid formaminifera, both groups produce calcareous tests via the intravesicular mineral precursors (Mg-rich amorphous calcium carbonate). These precursors are proposed to be supplied by endocytosed seawater and deposited in situ as mesocrystals formed at the site of new wall formation within the organic matrix. The authors did not observe the calcification of the needles within the transported vesicles, which challenges the previous model of miliolid mineralization. Although the authors argue that these two groups of foraminifera utilize the same calcification mechanism, they also suggest that these calcification pathways evolved independently in the Paleozoic.

Comments on the revised version

In my reply to the author's rebuttal letter, I will focus on one key point. The main observation supporting the author's conclusion, as expressed in the abstract, is:

"We found that both groups [i.e., rotaliids and miliolids, the latter documented in the reviewed paper] produced calcareous shells via the intravesicular formation of unstable mineral precursors (Mg-rich amorphous calcium carbonates) supplied by endocytosed seawater and deposited at the site of new wall formation within the organic matrix. Precipitation of high-Mg calcitic mesocrystals took place in situ and formed a dense, chaotic meshwork of needle-like crystallites."

In my review, I pointed out that there is no support for the existence of an intracellular, vesicular intermediate amorphous phase.

The authors replied:

"We used laser line 405 nm and multiphoton excitation to detect ACCs. These wavelengths (partly) permeate the shell to excite ACCs autofluorescence. The autofluorescence of the shells is present as well but not clearly visible in movie S4 as the fluorescence of ACCs is stronger. This may be related to the plane/section of the cell which is shown. The laser permeates the shell above the ACCs (short distance) but to excite the shell CaCO3 around foraminifera in the same three-dimensional section where ACCs are shown, the light must pass a thick CaCO3 area due to the three-dimensional structure of the foraminiferan shell. Therefore, the laser light intensity is reduced. In a revised version, a movie/image with reduced threshold is shown."

This reply does not address the reviewer's concerns. Detection of ACC with 405 nm excitation is not sufficient; many organic components can fluoresce under violet light excitation. For example, Delvene et al. (2002) (https://doi.org/10.18261/let.55.4.7) showed that "the Pleistocene and Jurassic microborings emit in the blue-yellow spectral region (420-600 nm) with a laser excitation of 405 nm, which coincides with the emission due to NADPH [nicotinamide adenine dinucleotide], FAD [flavin adenine dinucleotide], and riboflavin pigments characteristic of some cyanobacteria." Traditionally, in geological or biogenic calcium carbonate samples, Raman spectroscopic characterization of ACC and its magnesium content can be used (e.g., Wang, D., Hamm, L. M., Bodnar, R. J. & Dove, P. M. Raman spectroscopic characterization of the magnesium content in amorphous calcium carbonates. J. Raman Spectrosc. 43, 543-548 (2012); Perrin, J. et al. Raman characterization of synthetic magnesian calcites. Am. Mineral. 101, 2525-2538 (2016)). However, in biological, living-cell systems, Mehta et al. (2022) (doi: 10.1016/j.saa.2022.121262) successfully used FTIR spectroscopy to identify ACC by two characteristic FTIR vibrations at ca. 860 cm-1 and ca. 306 cm-1. Other methods such as STXM analyses at the C K-edge (Monteil et al. 2021, doi: 10.1038/s41396-020-00747-3) are also available. Because the core of the authors' interpretation (i.e., detection of ACC in vesicles) is not supported by hard evidence, the claim that the study represents a "paradigm shift" is far-fetched and the whole model is based on speculations. If the authors are able to unequivocally confirm the presence of ACC within the vesicles and its subsequent transformation into calcitic needles, the other problems noted in the paper will be relatively trivial.

---

## [Author Response]

The following is the authors’ response to the original reviews.

**Reviewer #1 (Public Review):**
Summary:The manuscript by Dubicka and co-workers on calcification in miliolid foraminifera presents an interesting piece of work. The study uses confocal and electron microscopy to show that the traditional picture of calcification in porcelaneous foraminifera is incorrect.Strengths:The authors present high-quality images and an original approach to a relatively solid (so I thought) model of calcification.Weaknesses:There are several major shortcomings. Despite the interesting subject and the wonderful images, the conclusions of this manuscript are simply not supported at all by the results. The fluorescent images may not have any relation to the process of calcification and should therefore not be part of this manuscript. The SEM images, however, do point to an outdated idea of miliolid calcification. I think the manuscript would be much stronger with the focus on the SEM images and with the speculation of the physiological processes greatly reduced.

We agree that fluorescence studies presented in the paper are not an unequivocal proof by itself for calcification model utilised by studied Miliolida species. However, fluorescence data combined with SEM studies, especially overlap of the elements that show autofluorescence upon excitation at 405 nm (emission 420–480 nm) and acidic vesicles marked by p_H-_sensitive LysoGlow84, may be a hint indicating ACC-bearing vesicles.

We will tone down the the physiological interpretation based on fluorescence studies in the revised version of the manuscript.

Nevertheless, we think that our fluorescent life-imaging experiments provides important observations in miliolida, which is scarce in the existing literature, and therefore are worth being presented as they might be very helpful in better understanding of full calcification model in the future.

**Reviewer #2 (Public Review):**
Summary:Dubicka et al. in their paper entitled " Biocalcification in porcelaneous foraminifera" suggest that in contrast to the traditionally claimed two different modes of test calcification by rotallid and porcelaneous miliolid formaminifera, both groups produce calcareous tests via the intravesicular mineral precursors (Mg-rich amorphous calcium carbonate). These precursors are proposed to be supplied by endocytosed seawater and deposited in situ as mesocrystals formed at the site of new wall formation within the organic matrix. The authors did not observe the calcification of the needles within the transported vesicles, which challenges the previous model of miliolid mineralization. Although the authors argue that these two groups of foraminifera utilize the same calcification mechanism, they also suggest that these calcification pathways evolved independently in the Paleozoic.

We do not argue that Miliolida and Rotallida utilize exactly the same calcification mechanism but the both groups use less divergent crystallization pathways, where mesocrystalline chamber walls are created by accumulating and assembling particles of pre-formed liquid amorphous mineral phase.

Strengths:The authors document various unknown aspects of calcification of Pseudolachlanella eburnea and elucidate some poorly explained phenomena (e.g., translucent properties of the freshly formed test) however there are several problematic observations/interpretations which in my opinion should be carefully addressed.Weaknesses:(1) The authors (line 122) suggest that "characteristic autofluorescence indicates the carbonate content of the vesicles (Fig. S2), which are considered to be Mg-ACCs (amorphous MgCaCO3) (Fig. 2, Movies S4 and S5)". Figure S2 which the authors refer to shows only broken sections of organic sheath at different stages of mineralization. Movie S4 shows that only in a few regions some vesicles exhibit red autofluorescence interpreted as Mg-ACC (S5 is missing but probably the authors were referring to S3). In their previous paper (Dubicka et al 2023: Heliyon), the authors used exactly the same methodology to suggest that these are intracellularly formed Mg-rich amorphous calcium carbonate particles that transform into a stable mineral phase in rotaliid Aphistegina lessonii. However, in Figure 1D (Dubicka et al 2023) the apparently carbonate-loaded vesicles show the same red autofluorescence as the test, whereas in their current paper, no evidence of autofluorescence of Mg-ACC grains accumulated within the "gel-like" organic matrix is given. The S3 and S4 movies show circulation of various fluorescing components, but no initial phase of test formation is observable (numerous mineral grains embedded within the o rganic matrix - Figures 3A and B - should be clearly observed also as autofluorescence of the whole layer). Thus the crucial argument supporting the calcification model (Figure 5) is missing.

This is correct that we did not observe the initial phase of test formation *in vivo*. Therefore, it is not our crucial argument supporting novel components of the new calcification model. We suspect that vesicles preparing and transporting Mg-ACC are produced way before their docking and deposition into the new wall, because such seawater vesicles were observed between the chamber formation stages (Goleń and Tyszka, 2024, personal communication based on independent experiments on a closely related miliolid taxon). It means that our *in vivo* experiments most likely represent a long, dynamic stage of vesicles formation via seawater endocytosis, their modification (incl. Mg-ACC formation) before the stage of exocytosis during the new chamber formation. Our crucial arguments supporting the calcification model come from the SEM imaging of the specimens fixed during chamber formation, as well as from the transparency of the new chamber wall during its progressive calcification.

There is no support for the following interpretation (lines 199-203) "The existence of intracellular, vesicular intermediate amorphous phase (Mg-ACC pools), which supply successive doses of carbonate material to shell production, was supported by autofluorescence (excitation at 405 nm; Fig. 2; Movies S3 and S4; see Dubicka et al., 2023) and a high content of Ca and Mg quantified from the area of cytoplasm by SEM-EDS analysis (Fig. S6)."

We used laser line 405nm and multiphoton excitaton to detect ACCs. These wavelengths (partly) permeate the shell to excite ACCs autofluorescence. The autofluorescence of the shells is present as well but not clearly visible in movieS4 as the fluorescence of ACCs is stronger. This may be related to the plane/section of the cell which is shown. The laser permeates the shell above the ACCs (short distance) but to excite the shell CaCO3 around foraminifera in the same three-dimensional section where ACCs are shown, the light must pass a thick CaCO3 area due to the three-dimensional structure of the foraminiferan shell. Therefore, the laser light intensity is reduced. In a revised version a movie/image with reduced threshold is shown.

**Author response image 1. sa3fig1:** Autofluorescence image of studied Miliolida species (exc. 405 nm) showing algal chlorophyll (blue) and CaCO3 (red), both ACC and calcite shell.

It would be very convenient if it was possible to visualize ACC by illumination with a blacklight, but there are very many organic molecules that have an autofluorescence excited by ~405 nm. One of the examples is NADH (Lee et al., 2015. Kor J Physiol Pharmac 19(4): 373-382), an omnipresent molecule in any cell (couldn't copy the appropriate picture here, but the reference has a figure with the em/exc spectra).

The paper of Lee et al. 2015 shows that the excitation spectrum of NADH is ending close to 400 nm. This means that NADH is not or only very weakly excitable at 405nm, what we used as the excitation laser line.

(2) The authors suggest that "no organic matter was detected between the needles of the porcelain structures (Figures 3E; 3E; S4C, and S5A)". Such a suggestion, which is highly unusual considering that biogenic minerals almost by definition contain various organic components, was made based only on FE-SEM observation. The authors should either provide clearcut evidence of the lack of organic matter (unlikely) or may suggest that intense calcium carbonate precipitation within organic matrix gel ultimately results in a decrease of the amount of the organic phase (but not its complete elimination), alike the pure calcium carbonate crystals are separated from the remaining liquid with impurities ("mother liquor"). On the other hand, if (249-250) "organic matrix involved in the biomineralization of foraminiferal shells may contain collagen-like networks", such "laminar" organization of the organic matrix may partly explain the arrangement of carbonate fibers parallel to the surface as observed in Fig. 3E1.

We agree with the reviewer that biogenic minerals should by definition contain some organic components. We just wrote that "no organic matter was detected between the needles of the porcelain structures” that means that we did not detect any organic structures based only on our FE-SEM observations. We will rephrase this part of the text to avoid further confusion.

(3) The author's observations indeed do not show the formation of individual skeletal crystallites within intracellular vesicles, however, do not explain either what is the structure of individual skeletal crystallites and how they are formed. Especially, what are the structures observed in polarized light (and interpreted as calcite crystallites) by De Nooijer et al. 2009? The author's explanation of the process (lines 213-216) is not particularly convincing "we suspect that the OM was removed from the test wall and recycled by the cell itself".

Thank you for this comment. We will do our best to supplement our explanations. We are aware about the structures observed in polarized light by De Nooijer et al. (2009). However, Goleń et al. (2022, Prostist; + 2 other citations) showed that organic polymers may also exhibit light polarization. Additional experimental studies are needed to separate these types of polarization. We will try to investigate this issue in our future research.

(4) The following passage (lines 296-304) which deals with the concept of mesocrystals is not supported by the authors' methodology or observations. The authors state that miliolid needles "assembled with calcite nanoparticles, are unique examples of biogenic mesocrystals (see Cölfen and Antonietti, 2005), forming distinct geometric shapes limited by planar crystalline faces" later in the same passage the authors say that "mesocrystals are common biogenic components in the skeletons of marine organisms" (are they thus unique or are they common)? It is my suggestion to completely eliminate this concept here until various crystallographic details of the miliolid test formation are well documented.

Our intension was to express that mesocrystals are common biogenic components in the skeletons of marine organisms however such a miliolid needles forming distinct geometric shapes limited by planar crystalline faces are unique.

**Reviewer #1 (Recommendations For The Authors):**
Below, I have summarized my main criticisms.(1) The movies S1-S4 do not indicate what is described. The videos show indeed seawater (S1), cell membranes (S2), and autofluorescence and acidic vesicles (S3 and S4). The presence of all these intracellular structures is not surprising: any eukaryotic cell will have those. The authors, however, claim that they participate in the process of calcification, which is simply not shown. One of the main arguments seems the presence of 'carbonate pools', in the caption these are even claimed to be 'Mg-ACC pools', but this is by no means revealed by an excitation of 405nm/ emission between 420 and 490 nm. It would be very convenient if it was possible to visualize ACC by illumination with a blacklight, but there are very many organic molecules that have an autofluorescence excited by ~405 nm. One of the examples is NADH (Lee et al., 2015. Kor J Physiol Pharmac 19(4): 373-382), an omnipresent molecule in any cell (couldn't copy the appropriate picture here, but the reference has a figure with the em/exc spectra).

The paper of Lee et al. 2015 shows that the excitation spectrum of NADH is ending close to 400 nm. This means that NADH is not or only very weakly excitable at 405nm, what we used as the excitation laser line.

The fluorescence by this excitation/ emission couple unlikely indicates the vesicles in which these foraminifera calcify. Therefore, most of the interpretation of the authors on what happens with the calcitic needles is not based on results but remains pure speculation.

The fluorescence autofluorescence upon excitation at 405 nm (emission 420–480 nm is typical for CaCO3 both for biocalcite and amorphous calcium carbonate, what was proven by laboratory synthesis of amorphous calcium carbonate Dubicka et al., in preparation).

(2) The results mention 'granules', which are the supposed Mg-ACC-containing vesicles, but the movies simply don't show any granules. Only fluorescence. Again, the results show a lot of vesicles with autofluorescence, but these are not necessarily related to calcification. Proof could be supplied by showing that the same fluorescent vesicles are 'used up' when the specimens under observation are making a new chamber, but until that is done, the fate of all these vesicles remains uncertain and once more, may not be involved in calcification at all.

We suspect that vesicles preparing and transporting Mg-ACC are produced way before their docking and deposition into the new wall, because such seawater vesicles were observed between the chamber formation stages (Goleń and Tyszka, 2024, personal communication based on independent experiments on a closely related miliolid taxon). It means that our *in vivo* experiments most likely represent a long, dynamic stage of vesicles formation via seawater endocytosis, their modification (incl. Mg-ACC formation) before the stage of exocytosis during the new chamber formation. Our crucial arguments supporting the calcification model come from the SEM imaging of the specimens fixed during chamber formation, as well as from the transparency of the new chamber wall during its progressive calcification.

(3) The Methods are unclear. How long were the foraminifers kept before being placed under the microscope? Were they fed with anything? This is important since the chlorophyll should not be from any food source. I didn't know that this foraminiferal species has photosynthetic symbionts: genera like Quinqueloculina don't. Is there any reference for this? Normally, I wouldn't care that much, but the authors find the presence of (facultative) symbionts important (lines 305-336). I am a bit suspicious about this since the only evidence for the presence of photosynthetic symbionts is because of the autofluorescence. As the authors said, commonly these miliolid species are regarded as symbiont-barren, so additional proof for these symbionts is necessary.

We agree that additional proof is needed for the presence of photosynthetic symbionts. We rephrased the manuscript accordingly.

(4) It is also unclear (Methods) at what stage the miliolids were photographed (Figure 3). How did chamber formation proceed, what was the timing of the photographs, etc. These pictures are to me the most interesting finding of this study, but need to be described much better.

All individuals of living foraminifera were fixed at the overall stage of chamber formation. However, every individual presents a complete set of successive steps (substages) of chamber wall calcification fixed at once. Fig. 3A and B present nearly the most proximal (youngest) part of the new chamber with a thick wall of calcite nanograins within a gel-like organic matrix. Fig. 3C and D present a bit more distal (intermediate) part of the calcified chamber. Fig. 3E shows the most distal part of the new chamber. This part is anchored to the older, underlying solid calcified chamber (not shown in this figure). All these steps are synchronous, however, represent gradual successive stages of calcification. The main text and Figs 4 and 5 explain this phenomenon in details.

There are many small issues with the text too. These include:Line 28/29: in many other groups, calcification is thought to be polyphyletic (e.g. sponges: Chombard et al., 1997. Biol Bull 193: 359-367).

Corrected

Line 29/30: there may be even more 'types of shells'. The first author has shown in earlier papers that nodosarids have a unique shell architecture. Spirillinids also seem to have their own way of calcification. It is unclear what is meant here by 'two contrasting models'.

By now there are known only two models of foraminiferal calcification. Lagenida biocalcification has not been studied.

Line 33: 'Both groups'? This paper only shows calcification in miliolids.

However, we refer to previous study.

Line 42: Perhaps, but there is no data on the pseudopodial network in this manuscript.

We refer to Angell, 1980 studies

Line 43: Likely, but that is not what this manuscript is showing.Line 42-44: The authors should make a choice and be clear. The point of this paper is that miliolids and rotalids calcify in ways that are actually not as different as they seemed previously. Still, they are said to have different 'chamber formation modes'. If they are calcifying in a similar way (which I think is not necessarily supported by the results), isn't calcification in these groups like variations on the same theme? How does this relate to the independent origins of calcification within these two groups?

Our intension is to show that Miliolida and Rotaliida utilize less divergent calcification pathways, following the recently discovered biomineralization principles.

Line 49-51: is this a well-established distinction? If so, please add a reference. If not: what is fundamentally different between B and C? Does only the size of the intracellular vesicle matter?

Rephrased

Line 60: please include a reference for the intracellular calcification by coccolithophores.

Added

Line 67: this is wrong. It is the alignment of the needles at the surface that makes them all reflect light in the same way and gives the shells a porcelaneous appearance. A close-up of the miliolid's shell surface shows this arrangement. Underneath this layer, the orientation of the needles is more random.

We referred to Johan Hohenegger papers.

Line 114: how else?Line 114-116: I don't see the relevance here. If seawater is taken up, the vesicle containing this seawater has to have a membrane around it. By definition. The text here ('These vesicles') suggests that Calcein and FM1-43 were combined (which they easily could have), but the methods describe that they are used successively.

Yes, we used two dyes separately.

Lines 122-130: I think the interpretation of this autofluorescence signal is wrong. Even if it was true, these lines belong to the Discussion.

This paragraph has been placed within discussion

Line 138: What are 'mobile clusters'? I don't see a relation between the location of the symbionts and the other vesicles (Figure 2).Line 147-148: How can an SEM image show the absence of organic matter?

We meant the absence of the gel-like OM visible in the previous stages of the chamber formation

Line 148: Should be 'Figs. 3E; 3E1; S4C'.

Corrected

Lines 143-150: this can be merged with the following paragraph.

Done

Lines 151-169: why is there no indication of the time? Figures 3 and 4 link the pictures in time to show the development of the growing chamber wall. However, neither here nor in the methods, is there any recording of the time after the beginning of chamber formation. Now, the images are linked (Figure 4) as if they were taken at regular intervals, but this is not documented.Lines 170-184: this should go to the Discussion.

Done

Line 193-195: this is likely, but not visible in Figure 1.

It was visible by optical microscopy and described by Angell, 1980

Line 199-201: I don't understand this: the fluorescent vesicles were not observed during chamber formation so any link between the SEM and CLSM scans remains pure speculation.Line 203-204: needed for what?

For better documentation of Miliolid ACC-bearing granules

Line 220: is this shown in any of the images?

Angell, 1980

Line 230: It sounds nice, but I don't think a 'paradigm shift' is appropriate here. However interesting and important foraminiferal biomineralization is, the authors show that the crystals of miliolids are likely formed differently than previously thought. If this is a 'paradigm shift', then most scientific findings are.

In our opinion this is definitely a shift of paradigm

Line 231: I don't think anyone suggested miliolids and coccolithophores share 'the same' pathway. They are shown (cocco's) and thought (miliolids) to secrete their calcite intracellularly.

Changed to similar, intracellular

Line 258: References should only be to peer-reviewed studies.Line 430: Burgers'

Corrected

**Reviewer #2 (Recommendations For The Authors):**
Please separate clearly the results (observations) from the discussion (interpretations): various interpretational/commentary phrases should be removed from the Results section to Discussion e.g., lines 124-130, 131-135.

Interpretation have been separated from results as suggested by Reviewer.

[line 49] " living cells have evolved three major skeleton crystallization pathways". I would rather say "organisms" not "cells" as the coordination of the calcification process in multicellular organisms clearly involves processes that are beyond the individual cell activity.

Corrected